# Comparative Analysis of Core Microbiome Assignments: Implications for Ecological Synthesis

Gordon F. Custer,[a,b,c,d] Maya Gans,[e] Linda T. A. van Diepen,[a,d] Francisco Dini-Andreote,[b,c] C. Alex Buerkle[d,e]

aDepartment of Ecosystem Science and Management, University of Wyoming, Laramie, Wyoming, USA
bDepartment of Plant Science, Pennsylvania State University, University Park, Pennsylvania, USA
cHuck Institutes of Life Sciences, Pennsylvania State University, University Park, Pennsylvania, USA
dProgram in Ecology, University of Wyoming, Laramie, Wyoming, USA
eDepartment of Botany, University of Wyoming, Laramie, Wyoming, USA

Gordon F. Custer and Maya Gans shared first authorship. Author order of Gordon F. Custer and Maya Gans was determined via a coin flip.

**ABSTRACT** The concept of a core microbiome has been broadly used to refer to the consistent presence of a set of taxa across multiple samples within a given habitat. The assignment of taxa to core microbiomes can be performed by several methods based on the abundance and occupancy (i.e., detection across samples) of individual taxa. These approaches have led to methodological inconsistencies, with direct implications for ecological interpretation. Here, we reviewed a set of methods most commonly used to infer core microbiomes in divergent systems. We applied these methods using large data sets and analyzed simulations to determine their accuracy in core microbiome assignments. Our results show that core taxa assignments vary significantly across methods and data set types, with occupancy-based methods most accurately defining true core membership. We also found the ability of these methods to accurately capture core assignments to be contingent on the distribution of taxon abundance and occupancy in the data set. Finally, we provide specific recommendations for further studies using core taxa assignments and discuss the need for unifying methodical approaches toward data processing to advance ecological synthesis.

**IMPORTANCE** Different methods are commonly used to assign core microbiome membership, leading to methodological inconsistencies across studies. In this study, we review a set of the most commonly used core microbiome assignment methods and compare their core assignments using both simulated and empirical data. We report inconsistent classifications from commonly applied core microbiome assignment methods. Furthermore, we demonstrate the implication that variable core assignments may have on downstream ecological interpretations. Although we still lack a standardized approach to core taxa assignments, our study provides a direction to properly test core assignment methods and offers advances in model parameterization and method choice across distinct data types.

**KEYWORDS** ecology, microbiome, simulation model, taxon abundance, taxon occupancy, abundance, prevalence

**Ad Hoc Peer Reviewer** Adriana Torres-Ballesteros

Address correspondence to Gordon F. Custer, gordon.custer91@gmail.com, or Maya Gans, jaffe.maya@gmail.com.
The authors declare no conflict of interest.

A core microbiome can be defined as a set of taxa that consistently occur within a given habitat type. This concept is explored by examining taxon abundances and occupancy across multiple samples and sometimes includes spatiotemporal variation (1). These common sets of microbial species within each habitat (e.g., human gastrointestinal tract, plant rhizosphere, soil, coral reef, etc.) are often assumed to be associated with the maintenance of baseline (eco)system functioning (1–8). As such, the

identification and conservation of core members across a variety of systems have become increasingly discussed in the literature, mostly in the context of a constantly changing climate and the maintenance of essential microbial-mediated (eco)system processes (2). With this, cataloging a core set of taxa associated with (and across) various systems has been a focus of many recent microbiome studies, which has resulted in a rapid increase in core microbiome assignments over the past 15 years (9).

The concept of a core microbiome has been applied to distinct systems using various sets of criteria (2–4, 6, 7, 9–14). For example, early efforts to describe the core microbiome of *Arabidopsis thaliana* identified the consistent enrichment of specific groups of endophytic Proteobacteria and Actinobacteria (10). Additionally, identifying microbial taxa that responded to plant or environmental stimuli was shown to be essential for follow-up plant-microbial manipulation studies (15, 16). Likewise, studies of activated sludge in wastewater treatment facilities have shown that core taxa are highly abundant and dynamically associated with ecosystem functioning (6). In addition, the use of core assignments in human microbiome studies has identified a link between gut microbiome assembly, nutrient acquisition, and obesity (3, 4). Here, these studies showed that instead of consistent taxon presence across 100% of subjects, the human gut hosts a stable functional core defined by the presence of common functional gene categories and metabolic pathways.

The use of idiosyncratic parameters and methods for core assignments across systems and studies can lead to divergent ecological interpretations and conclusions (9, 10, 17–19). Given the existence of these methodological inconsistencies, we here set out to (i) identify and organize results from the most common core assignment methods from the literature, (ii) evaluate the degree to which these methods result in consistent core assignments, (iii) benchmark the accuracy of each core assignment method using simulated data, and (iv) provide an example of how differences in core assignments can influence downstream statistical testing and affect ecological data interpretation. To do so, we first surveyed the literature for commonly used core microbiome assignment methods. We then used two publicly available data sets to test for consistency in core assignments among the methods identified in our literature search. Next, we tested core assignment methods using simulations to determine their accuracy across a range of plausible taxon distributions. Last, we provide statistical comparisons of core versus whole communities and enumerate implications for data interpretation and synthesis (i.e., identification of similar patterns in beta-diversity or in enumerating significant explanatory variables). Our findings are collectively summarized and discussed in line with the application of core assignments and appropriate ecological analysis. We advocate for the importance of considering robustness and methodological constraints toward advancing synthesis across core microbiome studies.

## RESULTS

**Comparison of core assignment methods.** To test for consistency in core taxa assignments we evaluated four distinct methods by using two large data sets: the plant rhizosphere and the human microbiome (Table 1). The four tested methods resulted in significant differences in core assignments (i.e., which taxa were assigned core membership and the number of assignments and percentage of total reads). These methods resulted in core assignments ranging from 1.21% to 9.42% of total taxa (Table 1), indicating a nearly 10-fold difference in the number of core taxa. When accounting for <10% of the total taxa, core assignments accounted for ~30% to 75% of the total reads (Fig. 1C and D). The evaluated methods mostly differed in core assignments when considering the coefficient of variance (CV) across treatment replicates (Fig. 2). That is, the abundance-based and the occupancy-based methods included highly abundant taxa regardless of the CV. However, the method based on abundance and occupancy selected only abundant taxa with a relatively low CV in the human microbiome data set (Fig. 2A), and abundant taxa regardless of the CV in the *Arabidopsis* data set (Fig. 2B). As such, this method appears to arbitrarily exclude taxa

**TABLE 1** Summary information for the two selected published data sets, including the number and percentage of operational taxa assigned to the core by each method tested[a]

| | Data set | |
|---|---|---|
| **Characteristic** | **Human Microbiome Project** | ***Arabidopsis thaliana* microbiome** |
| Total taxa | 11,752 | 14,890 |
| Total reads | 1,893,867 | 1,770,731 |
| Total samples | 319 | 288 |
| NCBI accession no. | HM16STR | ERP001384 |
| Sequencing platform | Illumina | 454 |
| **Method** | **Taxa assigned to core, *n* (%)** | |
| Abundance-based | 1,108 (9.42) | 1,245 (8.36) |
| Occupancy-based | 204 (1.73) | 1,134 (7.61) |
| Abundance and occupancy-based | 204 (1.73) | 907 (6.09) |
| Hard cutoffs of abundance and occupancy | 554 (4.71) | 181 (1.21) |
| Not assigned to the core by any method | 10,642 (90.55) | 13,590 (91.26) |
| Unique taxa assigned to the core by any method | 1,110 (9.44) | 1,300 (8.73) |

[a]The *Arabidopsis thaliana* data set was generated by Lundberg et al. (10) and only utilizes rhizosphere samples from the M21 site. The human microbiome data set was generated by the Human Microbiome Consortium (53) and includes only fecal samples.

with relatively high mean abundance and low CV, even though taxa with similar abundance and CV are included in the core. This is particularly evident in the analysis of the Human Microbiome Project data set (Fig. 2A).

The examined methods also showed the level of consistency in co-assignments (i.e., core assignment by multiple methods) to vary depending on the data set (Fig. 1A and B). The Human Microbiome Project data set yielded 176 consistent core assignments across the four methods (representing 15.85% of the total unique core assignments). Conversely, the *Arabidopsis* data set yielded 165 consistent core assignments in all four methods (representing 12.69% of the total unique core assignments). These common core assignments accounted for 1.49% and 1.1% of the total number of taxa in each data set, respectively. For the Human Microbiome Project data set, 376 taxa (33.87% of total unique core assignments) were assigned to the core by two methods, and 28 taxa (2.5% of total unique core assignments) were assigned by three methods. For the *Arabidopsis* data set, 188 taxa (14.4% of total unique core assignments) were assigned to the core by two methods and 742 taxa (57.1% of total unique core assignments) were assigned by three methods.

**Core microbiome and predictors of $\beta$-diversity.** To evaluate the degree to which core microbiomes result in similar patterns of $\beta$-diversity as the entire data sets, we compared the significance and explanatory power of variables in both data sets across the four methods. The complete Human Microbiome Project data set showed the categorical variables 'sex' and 'sequencing center' to be significant predictors of community variation (permutational multivariate analysis of variance [PERMANOVA], $P < 0.01$) (Table 2). This was true for both Bray-Curtis and Jaccard dissimilarity metrics. When the complete data set was reduced to examine only core taxa assignments, 'visit number,' 'sex,' and 'sequencing center' were found to be significant ($P < 0.05$) based on Bray-Curtis, and only 'sequencing center' was significant ($P < 0.001$) based on Jaccard. Results from $\beta$-diversity analysis for each of the four core-assignment methods were similar to the results obtained in the analysis of the complete data set for both dissimilarity indices, except for the abundance and occupancy-based method. In brief, while the abundance and occupancy-based method detected statistical significance for 'sequencing center', it also included 'visit number' as a significant predictor ($P < 0.05$) based on Bray-Curtis, although it was not significant for Jaccard ($P > 0.05$). Similarly, 'sex' as a predictor variable was found to be significant based on Bray-Curtis ($P < 0.01$), but not by Jaccard ($P > 0.05$).

Analysis of the significant predictors of $\beta$-diversity for the entire *Arabidopsis* data set revealed 'developmental stage' and 'genotype' to be significant ($P = 0.001$)

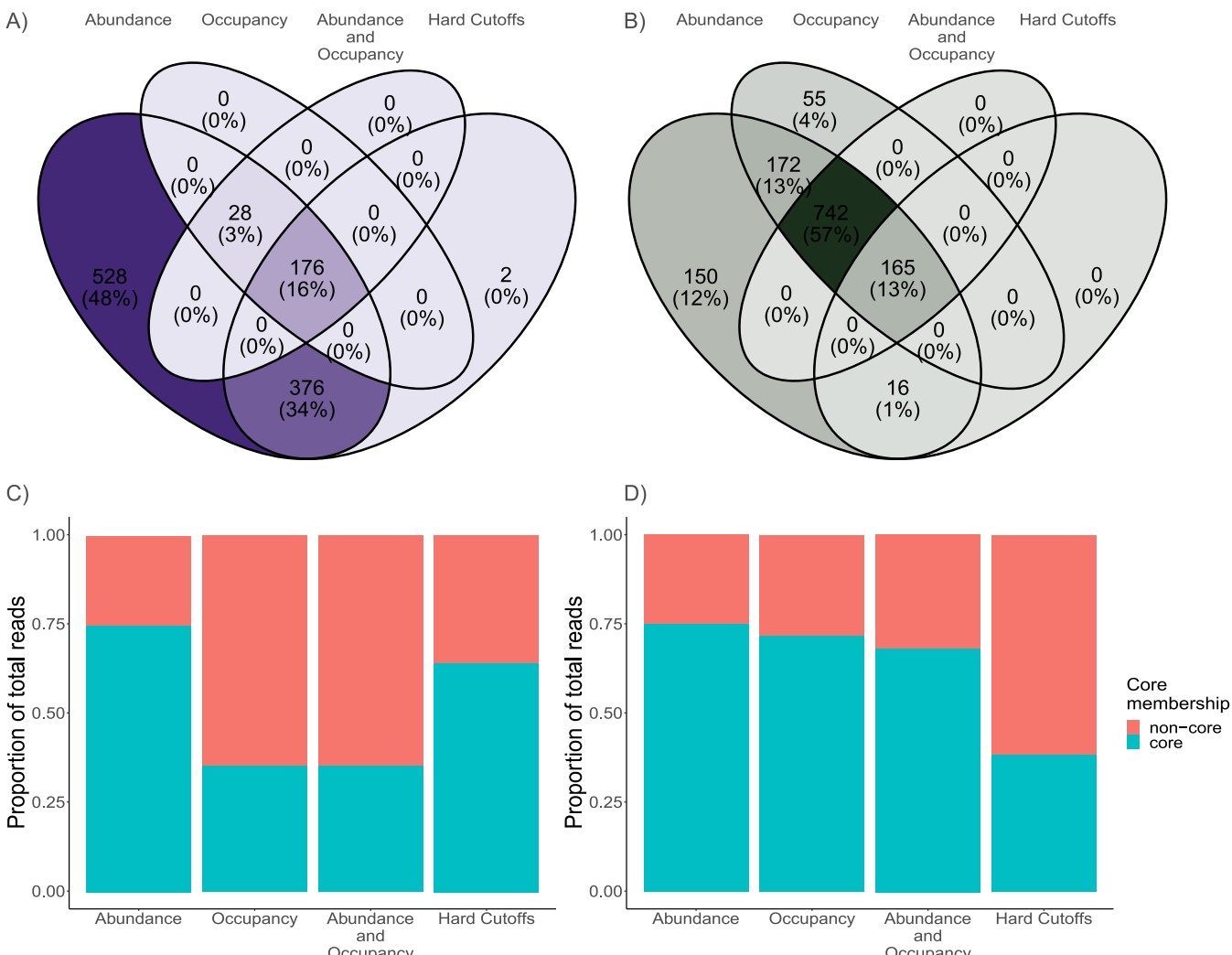

**FIG 1** Venn diagrams of co-assigned core taxa. (A) Core taxa co-assignments for the Human Microbiome Project data set. (B) Core taxa co-assignments for the *Arabidopsis* data set. Numbers in the overlapping regions represent the number of core assignments shared by that pair or combination of core assignment methods. Percentages represent the percentage of total core assignments. For example, in the *Arabidopsis* data set, the center of the Venn diagram shows that 165 core assignments were shared by all four methods. These 165 core assignments represent 13% of the total core assignments. (C) Proportion of reads accounted for by core and non-core members in the Human Microbiome Project data set for the four different core assignment methods. (D) Proportion of reads accounted for by core and non-core members in the *Arabidopsis* data set.

(Table 2). This was true for Bray-Curtis and Jaccard metrics. These results were corroborated when using only core taxa assigned by all four methods (*P* = 0.001) based on Bray-Curtis. However, Jaccard dissimilarity based on core taxa only significantly detected 'developmental stage' as a significant predictor (*P* = 0.001). In general, predictors of dissimilarity (e.g., *β*-diversity) using core assignments from each of the four methods produced similar results as the entire data set. The only exception was the hard cutoffs of the abundance and occupancy method. The results from this method were similar to those of the taxon table created from taxa co-assigned by all four methods, with 'developmental stage' being significant for both Bray-Curtis and Jaccard (*P* = 0.001), and 'genotype' being significant for Bray-Curtis (*P* = 0.001), but not for Jaccard (*P* = 0.148).

**Application of core assignment methods to simulated data.** We used simulation models to benchmark the accuracy of the four core assignment methods. Unlike real-world data sets, simulations allow for taxa with known core status to be assigned core membership. This permits rigorous assessment of the ability of each core method to accurately assign core membership. To benchmark the four methods, we used the net

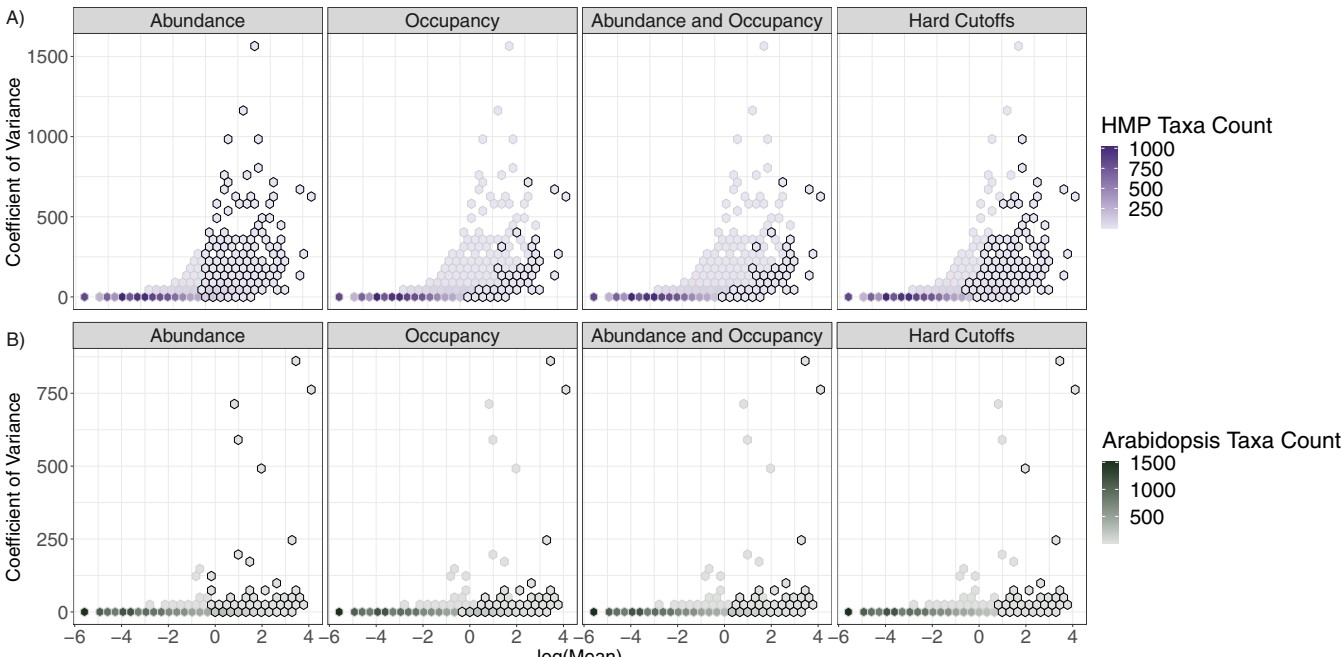

**FIG 2** Four core assignment methods identify different sets of core taxa from abundance data in a study of human microbiomes (top row) HMP, (Human Microbiome Project [52]) and (bottom row) *Arabidopsis thaliana*, (Lundberg et al. [10]). Additionally, the sets of core and non-core taxa do not exhibit categorically distinguishable abundances. The outlined bins denote whether taxa in the bin were included within the core. Fill color corresponds to taxa count.

assignment value to quantify differences in true positives (i.e., signal) (Fig. S1) and false positives (i.e., noise) in core taxa assignments (Fig. 3) (see Table 3 for definitions of all terms used). A net assignment value of 25 indicates maximum accuracy in core taxa assignment (with no erroneous assignments), while smaller values (including negative values) indicate progressively lower accuracy. In general, we found a large difference in the abundance of core and non-core taxa ($\pi_{core}/\pi_{non\text{-}core}$, with varying degrees of precision) to lead to greater accuracy in the identification of core taxa (Fig. 3, right side *x* axis: dark blue squares denote success, white and red indicate poor performance). This pattern was true for all methods, with larger numbers (i.e., most positive) being found toward the right side of the graphs regardless of the precision parameter ($\theta$). The occupancy-based method and the abundance and occupancy-based methods were able to accurately assign core taxa (i.e., net assignment value close to 25) in the upper right-hand side of the graphs (net assignment of >24). This occurred when core taxa were highly abundant and when there were small variations in abundance (i.e., consistently large differences in the abundances of core and non-core taxa). Conversely, the abundance-based method and hard cutoffs of abundance and occupancy performed poorly in terms of accuracy in core assignments under the conditions described above (net assignment values of ~925). These two methods produced the most accurate assignments in areas of the graphs corresponding to lower precision. In general, even their best assignments were not as accurate as the core assignments from the occupancy-based method and the abundance and occupancy-based method (i.e., both methods based on occupancy)—based on comparison of the same regions in the graphs (Fig. 3).

Even though core methods accurately assigned core membership in specific scenarios (e.g., large differences in abundance between core and non-core taxa and low variance), these same methods produced negative net assignment values in other scenarios. This resulted in an overestimation of core membership (Fig. 3). In brief, core inclusion was most severely overestimated by the hard cutoffs of the abundance and occupancy method and the abundance-based method in simulations with a low $\pi_{core}$

**TABLE 2** Variance explained by categorical predictors when applied to the entire data set and core assignments[a]

| | Categorical predictors | | | | | | | | | |
| | Human Microbiome Project data set | | | | | | Arabidopsis data set | | | |
| | Bray-Curtis | | | Binary | | | Bray-Curtis | | Binary | |
| Method | Visit no. | Patient sex | Sequencing center | Visit no. | Patient sex | Sequencing center | Developmental stage | Genotype | Developmental stage | Genotype |
|---|---|---|---|---|---|---|---|---|---|---|
| Full dataset | $P < 0.01$, $R^2 = 0.004$ | $P < 0.001$, $R^2 = 0.005$ | $P < 0.001$, $R^2 = 0.080$ | $P < 0.05$, $R^2 = 0.003$ | $P < 0.01$, $R^2 = 0.004$ | $P < 0.001$, $R^2 = 0.058$ | $P < 0.001$, $R^2 = 0.032$ | $P < 0.001$, $R^2 = 0.052$ | $P < 0.001$, $R^2 = 0.014$ | $P < 0.001$, $R^2 = 0.036$ |
| Co-assigned core by all methods | $P < 0.01$, $R^2 = 0.005$ | $P < 0.05$, $R^2 = 0.004$ | $P < 0.001$, $R^2 = 0.120$ | $P > 0.1$, $R^2 = 0.002^b$ | $P > 0.05$, $R^2 = 0.004^b$ | $P < 0.001$, $R^2 = 0.071$ | $P < 0.001$, $R^2 = 0.043$ | $P < 0.001$, $R^2 = 0.062$ | $P < 0.001$, $R^2 = 0.035$ | $P > 0.1$, $R^2 = 0.032^b$ |
| Abundance-based | $P < 0.05$, $R^2 = 0.004$ | $P < 0.01$, $R^2 = 0.005$ | $P < 0.001$, $R^2 = 0.090$ | $P > 0.1$, $R^2 = 0.003^b$ | $P < 0.01$, $R^2 = 0.005$ | $P < 0.001$, $R^2 = 0.069$ | $P < 0.001$, $R^2 = 0.039$ | $P < 0.001$, $R^2 = 0.059$ | $P < 0.001$, $R^2 = 0.024$ | $P < 0.001$, $R^2 = 0.045$ |
| Occupancy-based | $P < 0.05$, $R^2 = 0.006$ | $P < 0.05$, $R^2 = 0.005$ | $P < 0.001$, $R^2 = 0.119$ | $P > 0.1$, $R^2 = 0.003^b$ | $P > 0.05$, $R^2 = 0.004^b$ | $P < 0.001$, $R^2 = 0.069$ | $P < 0.001$, $R^2 = 0.038$ | $P < 0.001$, $R^2 = 0.059$ | $P < 0.001$, $R^2 = 0.020$ | $P < 0.001$, $R^2 = 0.045$ |
| Abundance and occupancy-based | $P < 0.01$, $R^2 = 0.006$ | $P < 0.05$, $R^2 = 0.005$ | $P < 0.001$, $R^2 = 0.080$ | $P > 0.1$, $R^2 = 0.003^b$ | $P > 0.05$, $R^2 = 0.004^b$ | $P < 0.001$, $R^2 = 0.069$ | $P < 0.001$, $R^2 = 0.039$ | $P < 0.001$, $R^2 = 0.061$ | $P < 0.001$, $R^2 = 0.021$ | $P < 0.001$, $R^2 = 0.047$ |
| Hard cutoffs of abundance and occupancy | $P < 0.05$, $R^2 = 0.004$ | $P < 0.01$, $R^2 = 0.005$ | $P < 0.001$, $R^2 = 0.094$ | $P > 0.5$, $R^2 = 0.002^b$ | $P < 0.05$, $R^2 = 0.004$ | $P < 0.001$, $R^2 = 0.069$ | $P < 0.001$, $R^2 = 0.043$ | $P < 0.001$, $R^2 = 0.062$ | $P < 0.001$, $R^2 = 0.033$ | $P > 0.1$, $R^2 = 0.033^b$ |

[a]$R^2$ values indicate the variance explained by a categorical predictor, and thus its importance.
[b]Significance of predictor differs between core assignment and the full data set.

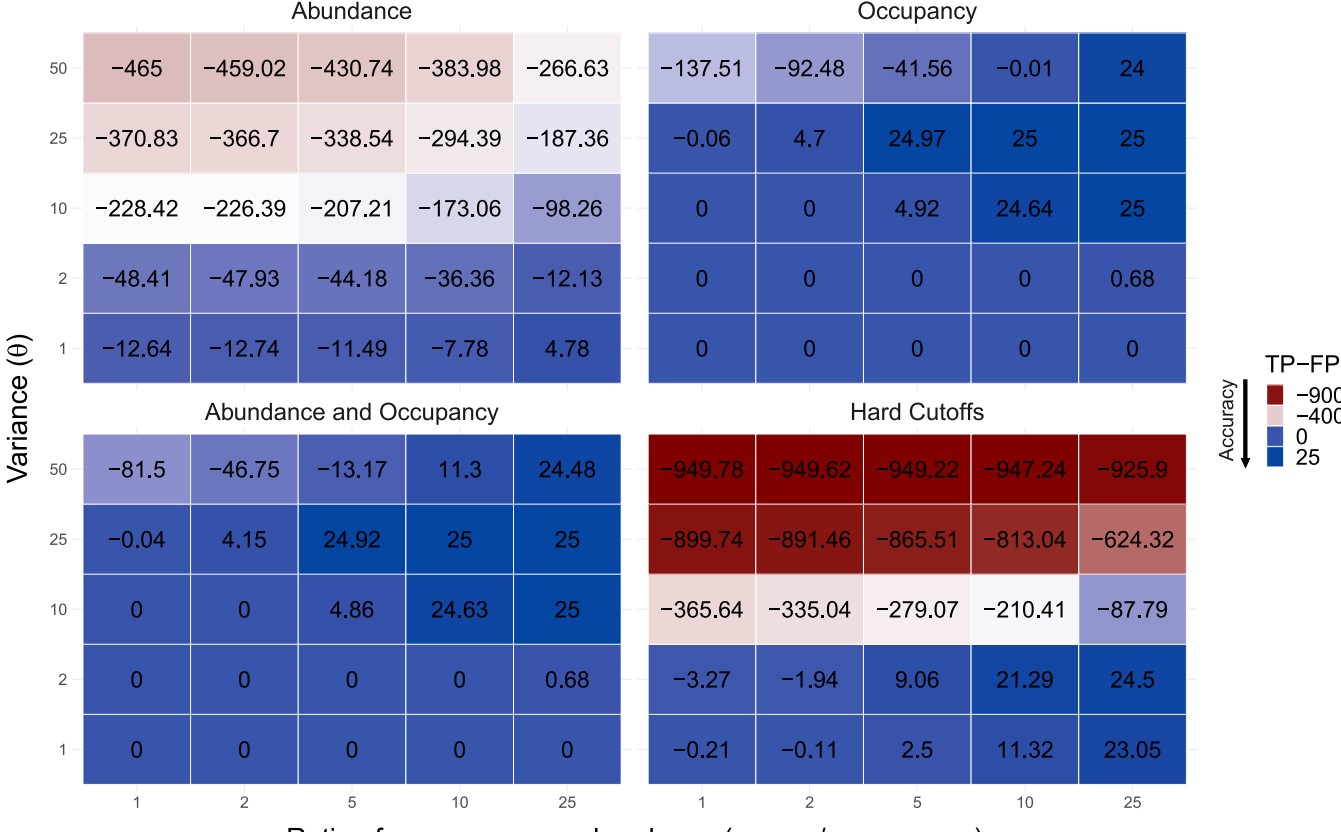

**FIG 3** Net assignment value for assignment of taxa to the core by four different core assignment methods (i.e., true positive [TP] to false positive [FP]). Each 5 × 5 heatmap represents 6,250 simulations. Each square within a heatmap represents 250 simulations at one of the possible 25 combinations of intensity ($\theta = 1$ indicates low precision in taxon abundance, with 2, 10, 25, and 50 corresponding to increasing precision in taxon frequency), and the ratio of the abundance of core to non-core taxa (with 1 being a simulation with no difference in the expected taxon frequencies, and 2, 5, 10, and 25 corresponding to greater differences in the frequency of core and non-core taxa). Here, we show the net assignment by method, an absolute value of true positives to false positives for each of the four core assignment methods. A positive net assignment value indicates better performance, while larger negative numbers indicate poorer assignment of core membership. Deep blue coloration indicates better performance of the core assignment method, with red indicating overinflation of the core.

to $\pi_{non-core}$ ratio and high precision (parameterized by $\theta$). The hard cutoffs of the abundance and occupancy method and the abundance-based method had 14 and 12 sets of simulations, respectively, with net assignment values lower than −200. This overestimation resulted in high false-positive rates (i.e., noise) in the simulations. In general, methods based on occupancy resulted in the assignment of smaller sets of core taxa and displayed the best net assignment values (i.e., higher accuracy). The occupancy-based method and the abundance and occupancy-based method each had six net assignment values of ∼24 or greater (i.e., nearly perfect assignment of core and non-core taxa).

## DISCUSSION

Increasing accuracy in core microbiome assignments across different systems and sample types remains a methodological and conceptual challenge. From a methodological standpoint, apart from issues arising from representative sampling efforts and technical inconsistencies (e.g., differences in sequencing platforms, sequencing depth, and sequencing quality), different core assignment methods and cutoff values have mostly been used in a user-defined manner. From a conceptual viewpoint, although the validity of assigning a core microbiome can be legitimately questioned (e.g., systems differ in microbiome stability or the importance of rare taxa on community functioning [20, 21]), it can be argued that the use of core assignments can provide opportunities for further hypothesis testing. For instance, core assignments may identify the most important

**TABLE 3** Definition of terminology used to describe simulations and core assignments

| Term | Definition or explanation |
|---|---|
| True positives/true core | Taxa simulated to have core abundance and occupancy that were assigned core membership by one of the assignment methods. |
| False positives/false core | Taxa simulated to have non-core abundance and occupancy but assigned core membership by one of the assignment methods. |
| True negatives | Taxa simulated with non-core abundance and occupancy that were not assigned core membership. |
| Net assignment value | Absolute difference between true positives and false positives. |
| Core:non-core abundance ratio ($\pi$) | Ratio of mean abundances between simulated core taxa and simulated non-core taxa. A larger value indicates that the core is more abundant relative to non-core members. For example, $\pi = 25$ indicates that the simulated mean abundances of core taxa are $25\times$ greater than the simulated mean abundance of non-core taxa. |
| Intensity parameter ($\theta$) | Affects variation in $\pi$ across replicates within a set of simulations. As the variance increases with a constant $\pi$, empty counts (e.g., 0's) are introduced to the count matrix. By holding $\pi$ constant within a set of simulations, the intensity parameter introduces variation in the occupancy of core and non-core taxa. |

set of taxa associated with the baseline functioning of a system, and further guide-targeted culturing or follow-up functional studies to test the validity of this assumption (22). Additionally, it can also provide initial evidence for potential metabolic interdependence among microbial species or between specific microbial taxa and their macro-organismal hosts (3, 4, 23). Here, we have provided a synthesis and benchmarking of the most common methods used to assign core membership across divergent microbial systems. To do so, we combined a literature search with simulation models to infer the accuracy of these different methods. Our analyses revealed varying levels of inconsistency across the most commonly used core assignment methods. Furthermore, we report that non-core taxa can significantly contribute to patterns of $\beta$-diversity and the importance of explanatory variables across systems. Although idiosyncrasies may continue to exist, our study provides information for advancing recommendations on core microbiome inferences.

**Inconsistences in core taxa assignment methods.** Overall, our results showed that the size of the core and the taxa assignments varied across the tested methods. This was shown to be further complicated by differences in the distribution of taxon abundances in each data set. These parameters are intrinsic to the data set type (e.g., habitat, host, etc.) and sample collection methods, and their importance for core assignments was validated by our simulations. Importantly, the size of the core varied across data sets even when comparing a single core assignment method. This implies that the most conservative core assignment method for one data set may not be the most conservative when applied to another.

Understanding the factors responsible for explaining variation in community composition is a common goal of microbiome studies. This is often associated with multivariate analysis aimed at establishing significant and strong correlations between community differences or turnover with environmental data (e.g., pH, nutrient concentrations, etc.). Previous attempts to better understand the contributions of core members to patterns of community composition have suggested that core members should account for a large proportion of similarity among samples. This is based on the fact that core taxa are more likely to be present and abundant across temporal and spatial replicates (12). By applying PERMANOVA testing to our data sets, we showed that core microbiomes have the potential to produce different statistical results and lead to alternate ecological interpretations compared to results obtained for the entire data set. When detected, these differences call attention to the importance of non-core members for explaining patterns in $\beta$-diversity and the statistical significance of predictor variables (12). For example, the significance of 'visit number' (a temporal effect) in the Human Microbiome Project was affected by the use of core assignments. In this case, the entire data set showed that 'visit number' had a significant effect on bacterial community composition (i.e., temporal variation). However, the application of PERMANOVA testing to core assignments showed the opposite and

suggested that a temporal effect is not significantly important. Given the selection of abundant or prevalent taxa by core assignment methods, variation in the statistical significance is likely driven by the inclusion or exclusion of rare members of the community. Most importantly, this might be tightly linked with the distribution of taxon abundances (e.g., evenness of taxa in the community) and the number of taxa (e.g., richness) in the data set. As such, these results point out that such differences in statistical associations can alter how researchers recognize the importance of predictors based on complete versus core community analyses, thus leading to alternate interpretations regarding temporal variation (24) or treatment effects.

**Benchmarking core taxa assignments via simulation models.** Our simulations considered true core taxa to be those that were 2 to 25 times more abundant than non-core taxa, with all core taxa simulated having higher abundances than their non-core counterparts. This provided core assignment methods the opportunity to accurately assign core taxa across a wide range of plausible community structures (i.e., variation in taxon distributions affected by $\pi_{core}/\pi_{non-core}$ and the precision parameter $\theta$). Unlike what may be observed in nature, the individual samples in our simulated data sets contained nearly identical taxon distributions, which offered the best-case scenario for all core assignment methods. Nevertheless, we found remarkable inconsistencies across methods in this scenario, which directly reflect the method performances. Most interestingly, we found that even the best-performing methods (i.e., those which most accurately assigned core membership) were affected by differences in abundance between core and non-core taxa (e.g., differential performance along the abundance axis, as shown in Fig. 3). For example, despite the overall higher accuracy of occupancy-based methods, as validated by the simulation models, these methods produced inaccurate assignments of core taxa in data sets displaying only small differences in abundance between core and non-core taxa. This often leads to an overinflation of the core size due to erroneous assignments. This pattern indicates that both the abundance and occupancy of taxa were important for accurately determining core membership in our simulations. Other methods, such as the hard cutoffs of abundance and occupancy and the abundance-based methods, revealed overinflated cores across a wider range of simulations. While core inclusion for simulations with high precision was severely overestimated, there were instances (i.e., low precision and a large difference in core and non-core abundances) in which the hard cutoff method assigned accurate cores. This indicates that the underlying distribution of taxon abundances and occupancy determines the accuracy of a core assignment method.

**Core microbiomes: challenges and limitations.** The use of core assignments based on operational taxonomic units/amplicon sequence variants (OTUs/ASVs) (i.e., binned sequences) has received pushback, and core assignments based on function have been suggested as an alternative (25). Despite this, core assignment methods that rely on binned sequences are more commonly used (9). This mostly occurs due to the cost and precision of data generation and annotation (i.e., the cost and resources required for marker gene versus metagenomic sequencing and processing), and the applicability of macroecological frameworks for identifying core taxa (i.e., thresholds in abundance and occupancy) (12). Further challenges stem from the vastly different definitions used to define core taxa and the subsequent variation in assignment methods. Core assignment methods based upon abundance emphasize taxa that resemble *r*-selected organisms, which are characterized by high growth rates and thus observed in high numbers. Those that do not meet an abundance threshold are generally excluded. On the other hand, occupancy-based assignment methods rely on adequate sequencing depth to accurately depict the presence (or absence) of a taxon in a set of samples. In situations where many samples are multiplexed for sequencing and reads are divided among samples, each sample receives a smaller proportion of reads and thus, less abundant individuals may not be detected. This can lead to a core that may not accurately represent the sampled communities. Furthermore, it is worth mentioning that sequence read quality and depth can also affect I/ASV binning and detection, respectively, with direct implications for core assignments based on abundance and occupancy.

Given that analyses of microbial communities are frequently based on counts of thousands of taxa across many samples (i.e., high dimensionality), the appeal of dimension reduction is apparent. In this sense, dimension reduction can decrease noise and variation among samples and resolve the strongest patterns in the data (26). Because many statistical methods lack power when applied to high dimensional data (26), focusing on a subset of the most common individuals can further enhance statistical power (27). From a conceptual standpoint, the practice of focusing on a core set of taxa is supported if the ecological functions of interest are driven by and associated with variation in the core set of taxa (i.e., those that are abundant, prevalent, or both) (12). There are certainly examples (28) and even theories (e.g., the mass-ratio hypothesis [29]) of ecological processes being tied to variation in the abundance of a small number of relatively common taxa. However, beyond statistical considerations, the contributions of less common individuals are becoming increasingly recognized (21, 30–32), with variation in ecological, especially microbially driven, processes being associated with rare taxa (e.g., sulfur reduction, nitrification, and methanogenesis). The importance of rare taxa demonstrates a potential pitfall for the application of core assignment methods and the use of core microbiomes for subsequent ecological interpretation (21, 31–33). In addition, dichotomous assignment to core and non-core ignores situations in which taxa, both common and rare organisms, form networks to produce complex trophic cascades (34, 35). Last, some studies have indicated that certain habitats are not occupied by a consistent, core set of taxa but instead mostly by transients (1, 18), thus calling into question the broad use of the core concept (36–38).

**Recommendations and outlook.** Given the idiosyncrasies in core assignments across methods, we first and foremost suggest that researchers using core assignments grasp the implications of 'discarding' a large subset of their data—empirically and in terms of interpretation. Our results revealed methodological inconsistencies and disagreement in follow-up analysis considering either the whole data set or their respective core communities. This has the potential to produce different interpretations regarding the importance of covariates when explaining variation in communities. Alternatively, researchers should strive to use all of the available abundance data, including rare taxa, in statistical procedures and rely on model-based methods to recognize groups of taxa that differ in abundance (39, 40). These include standard methods for multivariate analysis and dimension reduction (41, 42). Additionally, specialized methods for differential abundance analysis exist, including the Dirichlet multinomial (40, 43–45) and related methods which model the relative abundance of all taxa (46–50).

In summary, our application of core assignment methods to simulated and published data sets demonstrated significant inconsistent classifications from commonly applied criteria for determining core membership. Although we still lack a standardized approach to core taxa assignments and may even debate its ecological appropriateness across systems, our study provides a direction to properly test core assignment methods. Here, we offer advances in model parameterization and method choice across distinct data set types. Given the importance of comparisons across studies within similar systems (for instance, to promote ecological synthesis), equating core communities assigned by different methods may not be appropriate. Instead, the use of probabilistic tools to model the distribution and responses of microbial taxa is more likely to yield insights into microbial ecology (12). If core assignments are necessary, our results suggest that the methods based upon occupancy are the most accurate, regardless of differences in abundance between core and non-core taxa. Additionally, when variation in $\beta$-diversity and predictor variables are similarly detected using the whole community data set and the core subset, follow-up ecological interpretations and hypothesis-driven studies can more fundamentally be linked to core membership taxa. Finally, we emphasize the need for researchers to use core assignments to thoroughly detail their methodology. This represents a critical step in facilitating comparisons across studies toward consistency in ecological interpretation and developing synthesis of core microbiome outcomes in divergent systems.

**TABLE 4** Five commonly employed core methods along with descriptions and the number of publications using them found within Web of Science, accessed April 2018[a]

| Core method | Description | No. of publications |
|---|---|---|
| Abundance-based | Accounts for read depth (a portion of the top 75% of total reads); abundance | 6 |
| Occupancy-based | Accounts for sample size (present in >50% of samples); occupancy | 24 |
| Abundance and occupancy-based | Accounts for both sample size and read depth (present in >50% of samples with minimum abundance >0.02% of total reads); abundance and occupancy | 3 |
| Hard cutoffs of abundance and occupancy | Uses a specific no. of samples or reads (present in 5 samples with 25 total reads); abundance and occupancy | 4 |
| Venn diagram (e.g., stringent occupancy)[a] | Present in all subsets of samples | 8 |

[a]The Venn diagram method was not utilized in our analysis due to its similarity to the proportion of replicates method.

## MATERIALS AND METHODS

**Literature review and synthesis.** We conducted a literature review using Web of Science (April 2018) to identify and define the most common methods applied for core microbiome assignments. We limited our search to articles containing the terms 'core' and 'microbiome' within the title or abstract. This resulted in a total of 1,034 peer-reviewed articles. We selected publications from 2008 to 2018 and ordered search results by the number of citations. Then, we narrowed the data set to the top 200 most-cited articles. Within these 200 articles, only 45 publications sufficiently detailed their methods for assigning core taxa membership. We subdivided these methods into five categories based on methodological similarities (Table 4). Overall, this initial survey provided a representation and summary of core assignment methodologies used in contemporary analyses. We acknowledge that this survey does not include every method used for core assignments but instead represents the most commonly used to date. We identified five main groups of core assignment methods: abundance, occupancy, abundance and occupancy, hard cutoffs of abundance and occupancy, and Venn diagrams. Here, we reviewed and tested these methods (except for the Venn diagram, as it can be considered a stringent version of the occupancy-based method) using different data sets and simulation models.

**Abundance-based.** This method assigns core membership to the most abundant taxa in a data set. In our analysis, all taxa were ranked from highest to lowest based on their relative abundances. Then, core membership was assigned to all taxa accounting for a portion of the first 75% of total reads. This method has been previously adopted in plant ecology (51) and explicitly considers the relative abundance of individual taxa. In microbiome studies, this method is strongly affected by sample sequencing depth. The potential drawbacks of this method include the fact that a taxon does not have to be consistently present across multiple samples to be assigned core membership. Spurious core assignments can arise when a taxon is highly abundant in one subset of samples and absent in others. For our analysis, we set the cutoff at 75% of sequence reads.

**Occupancy-based.** This method assigns core membership to taxa occurring in a user-defined proportion of samples within a given treatment, habitat type, or observational category. The occurrence threshold varies across studies but is usually greater than 50% of samples and, in some cases, as high as 100%. In our analyses, core membership was assigned to taxa consistently detected in the majority of samples (e.g., occupancy > 50%). This method accounts for the number of samples collected using a function of occupancy as the sole determinant for core assignment. This method has the potential to include rare taxa because it is independent of individual taxon abundance. Drawbacks of this method include instances in which studies are based on a limited number of samples and the fact that it fails to consider the importance of abundance for core membership assignment. Additionally, this method is affected by sequencing depth because rare individuals may drop below detection limits with limited sequencing.

**Abundance and occupancy-based.** This method assigns core membership based on the abundance (i.e., accounting for a minimum number of reads in a data set) and occupancy of taxa (i.e., found in a minimum proportion of samples) (7). As such, this method accounts for both sample size and sampling effort. We found that different studies in the literature used variable occupancy (i.e., the proportion of samples) and abundance (i.e., the proportion of reads) thresholds. For our analysis, we used an occupancy greater than 50% and a total abundance greater than 0.02% across all samples (adapted from Callahan et al. [14]). One potential drawback of this method includes the high probability of variable core membership when different user-defined cutoffs are used.

**Hard cutoffs of abundance and occupancy.** This method also assigns core taxa membership based on occupancy and abundance. However, it differs from the abundance and occupancy-based method because it *a priori* assigns an absolute value for the cutoff. While this method is analogous to abundance and occupancy-based methods, we included it in our analyses to better understand the implications of using a hard cutoff across studies with different sample sizes and sequencing depths. For our analysis, we set the following cutoffs: taxon presence (i.e., occupancy) in 5 samples with at least 25 total reads across all samples (adapted from Lundberg et al. [10]). Potential drawbacks of this method include the fact that cutoffs do not change based on the size of the experiment or sequencing depth. As such, failing to account for sequencing depth and an increasing number of replicates has the potential to inflate

core microbiome assignments. This is particularly important when cutoffs account for a very small or large proportion of the total effort, as is the case of studies with small or large sample sizes, respectively.

**Applying different methods of core assignment.** We used two large data sets to examine consistency in core microbiome assignments with the four methods described above. The two large data sets were obtained from highly cited publications and represent complex microbial systems, i.e., the plant rhizosphere and the human microbiome (Table 1). In particular, we used (i) the rhizosphere and site M21 subset of the final rarified operational taxon table from the *Arabidopsis thaliana* root microbiome project (10), and (ii) the fecal subset of samples from the Human Microbiome Project (52). For both published data sets, we obtained post-processed OTU tables based on 97% nucleotide identity. Additional information on sequence data processing and analysis has been previously provided (10, 52).

To examine the distribution of core assignments and their non-core counterparts, we created bivariate plots by plotting the log-transformed mean taxon abundance and the coefficient of variance. This was done to assess whether core assignment methods identify thresholds in abundance and coefficient of variation (a measure of dispersion relative to the mean) between core and non-core taxa.

**Core microbiomes and categorical predictors of $\beta$-diversity.** Because core microbiomes can represent consistent sets of taxa across multiple samples and not necessarily consistent abundances, we tested whether core assignments produce similar results in identifying the best predictor variables for explaining patterns of community $\beta$-diversity. We used both weighted and unweighted community distance metrics (i.e., Bray-Curtis and Jaccard) to determine distances between samples using the core and whole data sets. The significance of predictors for explaining patterns in $\beta$-diversity was determined and compared using PERMANOVA (adonis) with 1,000 permutations using the vegan package (41). For the Human Microbiome Project data set, we examined three categorical predictors: patient visit number (1st, 2nd, or 3rd), subject sex (male versus female), and sequencing center ($n = 12$ different centers). For the *Arabidopsis* data set, we examined two categorical predictors: plant developmental stage (young versus old) and genotype (a total of 9 different genotypes).

**Simulation models to test core assignments.** To test the accuracy of core microbiome assignments, we ran a series of simulation models in R v3.4.2 (53). Given that core assignments are determined from ecological count data with unknown underlying taxon distributions, it is impossible to know which method produces the most accurate and precise core assignment (i.e., *a priori* knowledge of which taxa are truly core members). To address this challenge, we simulated 250 taxon tables for each of the 25 possible combinations of (i) five levels of magnitude of difference in abundance ($\pi$) of core versus non-core taxa (represented as the $\pi_{core}/\pi_{non-core}$, ranging from $1\times$ to $25\times$ difference), and (ii) five levels of variance in abundance ($\pi_{core}$ to $\pi_{non-core}$) among replicates (quantified by an intensity parameter, $\theta$, ranging from 1 to 50). This resulted in a total of 6,250 unique simulated taxon tables.

Each simulation of taxon abundances involved random draws from a Dirichlet distribution parameterized by the expected frequencies of all taxa ($\Sigma\pi_i = 1$, with 25 taxa parameterized by $\pi_{core}$ and 975 by $\pi_{non-core}$). This was based on a single intensity parameter ($\theta$) that affects the precision of taxon abundances (i.e., scales the variance around expected taxon abundance defined by $\pi_{core}$ and $\pi_{non-core}$). Across sets of simulations, we varied the relative abundance of core and non-core taxa ($\pi_{core}/\pi_{non-core}$), with $\pi_{core}/\pi_{non-core} = 1$ corresponding to a community that lacks a true core and all taxa having equal expected relative abundances. On the other end, $\pi_{core}/\pi_{non-core} = 25$ simulated a data set in which core taxa had a relative abundance 25 times greater than those of non-core taxa. Further, we used the intensity parameter $\theta$ to set the precision of taxon abundances across replicates for a given set of expected frequencies ($\pi$). Here, a $\theta$ of 50 corresponded to high precision and low variance in taxon relative abundances among replicates, and a $\theta$ of 1 denoted low precision and a large variance in taxon relative abundances, thus affecting occupancy across simulated samples. All simulations with a $\pi_{core}/\pi_{non-core}$ of >1 had 25 core taxa and the remaining 975 as non-core taxa. Consequently, up to 25 taxa could be detected as true core taxa (i.e., true positives) and 975 taxa as false core members (i.e., false positives) or true non-core members (i.e., true negatives). In brief, our simulations identified true core taxa (i.e., taxa assigned core membership and simulated to have core abundance and occupancy), false positives (taxa assigned to core membership but simulated to have non-core abundance and occupancy), and true negatives (i.e., taxa assigned non-core membership and simulated to have non-core abundance and occupancy). A simulation's random draw from the Dirichlet distribution yielded a vector of sample proportions for each of 1,000 taxa ($P[p_1, p_2, \ldots, p_{1,000}] \mid \pi_{core}, \pi_{non-core}, \theta$), to which the four criteria for core membership were applied.

**Quantification of accuracy in core microbiome assignments using simulation models.** The ability of each method to accurately identify the true core was assessed using simulated taxon tables and the following metrics: proportion of true positives (signal), proportion of false positives (noise), and net assignment value (signal-noise). The proportion of true positives represents the proportion of known core taxa, regardless of the number of false negatives. This is expressed as the probability of a true core taxon being properly assigned (e.g., a true positive proportion of 1 indicates accurate assignments). The proportion of false positives represents the proportion of non-core taxa classified as core and represents the probability of a non-core taxon being assigned as a member of the core (e.g., a false positive proportion of 0 indicates accurate assignments). The net assignment value represents the difference between the absolute number of true positives and the number of false positives. A net assignment value of 25 represents perfect classification. A net assignment value of $-975$ indicates that all non-core taxa were ill-assigned to the core with no true positive classifications. That is, larger negative numbers indicate highly inflated core assignments. This metric can be interpreted as the difference in signal and noise.

**Data availability.** We wrote the R package 'CoreMicro' to facilitate reproducibility in core microbiome assignments. This package is available at github.com/mayagans/coremicro and includes functions that

accept a taxon table as the input. It can be used to generate plots and tables of core inclusion by each of the methods tested in this study. The functions can be easily customized to accept different thresholds for core inclusion. This functionalized approach facilitates reproducibility and comparative analysis between methods, thus representing a platform for unifying core microbiome assignments. In addition, the package contains all data and code used to produce the simulations in this study. Full taxon tables and metadata files of the *Arabidopsis* and Human Microbiome Project data sets can be found at https://doi.org/10.5281/zenodo.7544753.

## SUPPLEMENTAL MATERIAL

Supplemental material is available online only.

**FIG S1**, EPS file, 2.6 MB.

## ACKNOWLEDGMENTS

We thank Reilly Dibner, Joshua Harrison, and Paul Ayayee for providing feedback on early drafts of the manuscript. Additionally, we thank the reviewers for providing constructive feedback and suggestion that greatly improved the manuscript's quality and readability.

This research was supported by the Microbial Ecology Collaborative with funding from NSF award no. EPS-1655726.

M.G. and G.F.C. conceived the ideas presented. M.G. conducted the literature review. M.G., G.F.C., and C.A.B. wrote code for analyses, figure creation, and simulations. M.G., G.F.C., F.D.-A., L.T.A.v.D., and C.A.B. developed and edited the manuscript. CoreMicro R package was developed by M.G. and G.F.C.

We declare that we have no competing interests.

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
