## [Reviewer comments · mSystems]

Comparative analysis of core microbiome assignments: implications for ecological synthesis

Gordon Custer, Maya Gans, Francisco Dini Andreote, Linda van Diepen, and C. Buerkle

Corresponding Author(s): Gordon Custer, The Pennsylvania State University

Review Timeline:

Submission Date:	November 1, 2022
Editorial Decision:	December 11, 2022
Revision Received:	January 3, 2023
Accepted:	January 4, 2023

Editor: Ashley Shade

Reviewer(s): Disclosure of reviewer identity is with reference to reviewer comments included in decision letter(s). The following individuals involved in review of your submission have agreed to reveal their identity: Adriana Torres-Ballesteros (Reviewer #3)

Transaction Report:

DOI: <https://doi.org/10.1128/msystems.01066-22>

December 11, 2022

Dr. Gordon Fritz Custer
The Pennsylvania State University
University Park, PA 16802

Re: mSystems01066-22 (Comparative analysis of core microbiome assignments: implications for ecological synthesis)

Dear Dr. Gordon Fritz Custer:

Thank you for submitting your manuscript to mSystems. We have completed our review and I am pleased to inform you that, in principle, we expect to accept it for publication in mSystems. However, acceptance will not be final until you have adequately addressed the reviewer comments.

Preparing Revision Guidelines

Sincerely,

Ashley Shade

Editor, mSystems

Journals Department
Reviewer comments:

Reviewer #3 (Comments for the Author):

General Comments

This paper summarizes the most used methods to assign core microbiome, a concept that now has become popular in microbial ecology. I really liked that they build an R package where you can find the most used methods and algorithms to define a "core microbiome". The simulations gave this work a strong support to develop the discussion of results.

Compared to the first version of the paper, the message that the authors want to communicate is now clearer. However, I believe the writing can be improved to make the paper lighter and easier to follow. For example, there is an excessive use of articles and use of necessary words that make sentences too long.

Answers to reviewer #1

Answers to reviewers' comments related to taxonomy assignment gave clarity about the focus of the "core microbiome" definition in this work.

Answers to reviewer #2

Compared to the first paper, now the logic structure in the manuscript is clearer

The terms of "true or "accurate" core where clarified.

The extra analysis performed to address concerns about meaningful differences in core membership, give evidence and statistical support to show the reader the significant differences when using each method.

Could you change the names you used to classify the core membership assignment methods?

Instead of summation of sequence reads, you can use "Abundance" and instead of proportion of replicates, you can use "Occurrence". For methods including both you can name it "Abundance-occurrence". This way will be easier to compare with other papers discussion about the core membership assignment methods.

Reviewer #4 (Comments for the Author):

I was not one of the original reviewers, but I have read the revised manuscript and the author responses to the original comments. In general, I agree with the previous reviewers about the expected utility of the R package for microbiome scientists who want to explore different core definitions on their datasets, and with the utility of simulations in better exploring core definitions. I also agree with the previous' reviewers sentiment that the readers may be left with a "now what?" feeling at the end of the work, as the authors conclude that these various definitions may not be useful. Ultimately, I assess that the authors have addressed the original major comments while leaving open to the readers to decide whether and how they want to apply a core definition based on their study's scientific question.

I offer two additional comments (which do not require additional analysis).

First, the authors describe the importance of the separate factors of abundance and prevalence. There was a 2019 piece in Current Opinion in Microbiology that directly considered both types of factors in determining a core and then also partitioned the contribution of the core to the beta diversity explained by the "core" subset as compared the whole community. This prior piece has a similar philosophy as the current piece. While the prior work is referenced among other review-type articles early in the introduction, it is not discussed in the context of this new study. However, the 2019 piece introduces the idea that the contribution of the core members to beta diversity is a metric of the "success" of the core definition applied to a study, as core members would ultimately be the taxa driving the major ecological patterns, which is now reinforced in the current study. In this sense, the 2019 piece can be used as precedent to motivate the consideration of the core contribution to beta diversity as an output of ecological insight. This discussion of the 2019 reference would potentially fit into the paragraph starting line 339 about predictors of beta diversity.

Reference: Shade, A., & Stopnisek, N. (2019). Abundance-occupancy distributions to prioritize plant core microbiome membership. Current opinion in microbiology, 49, 50-58.

Second, in the simulations the authors vary magnitude of difference and the variance of the abundances of the taxa in the simulated communities and define their "true core" by 1x- 25x differences in the abundance. If I understand this correctly, all true core taxa will have higher abundance than non-core taxa in the simulations. In addition, the prevalence parameter is considered as introduction of noise (precision) across replicates (as an outcome of potential incomplete observation?) and not as directly as their distribution e.g. over space or time/across samples collected in a study. Thus, the simulation seems to bias towards the abundance factor in its exploration of core definitions and uses a replicate-based definition of prevalence/precision rather than a dataset-wide definition. Given this, perhaps it is not surprising that the results confirm the importance of the differential abundances between true core and non-core (e.g., Line 282 "In general, we found a large difference in the abundance of core and non-core taxa ($\pi_{core}/\pi_{non-core}$, with varying degrees of precision) to lead to greater accuracy in the identification of core

taxa". This point - that the simulations create that core taxa by definition have higher abundance than non-core - should be elevated and discussed so that readers can more clearly understand and evaluate. This comment echoes a previous reviewer's request to clarify the simulations to "lay" readers so that they can understand the assumptions and their consequences.

General Comments

This paper summarizes the most used methods to assign core microbiome, a concept that now has **Error! Hyperlink reference not valid.** become popular in microbial ecology. I really liked that they build an R package where you can find the most used methods and algorithms to define a “core microbiome”. The simulations gave this work a strong support to develop the discussion of results.

Compared to the first version of the paper, the message that the authors want to communicate is now clearer. However, I believe the writing can be improved to make the paper lighter and easier to follow. For example, there is an excessive use of articles and use of necessary words that make sentences too long.

Answers to reviewer #1

Answers to reviewers' comments related to taxonomy assignment gave clarity about the focus of the “core microbiome” definition in this work.

Answers to reviewer #2

Compared to the first paper, now the logic structure in the manuscript is clearer

The terms of “true or “accurate” core where clarified.

The extra analysis performed to address concerns about meaningful differences in core membership, give evidence and statistical support to show the reader the significant differences when using each method.

Could you change the names you used to classify the core membership assignment methods?

Instead of summation of sequence reads, you can use “Abundance” and instead of proportion of replicates, you can use “Occurrence”. For methods including both you can name it “Abundance-occurrence”. This way will be easier to compare with other papers discussion about the core membership assignment methods.

Reviewer comments:

Reviewer #3 (Comments for the Author):

General Comments

This paper summarizes the most used methods to assign core microbiome, a concept that now has become popular in microbial ecology. I really liked that they build an R package where you can find the most used methods and algorithms to define a "core microbiome". The simulations gave this work a strong support to develop the discussion of results.

Compared to the first version of the paper, the message that the authors want to communicate is now clearer. However, I believe the writing can be improved to make the paper lighter and easier to follow. For example, there is an excessive use of articles and use of necessary words that make sentences too long.

Thank you for the positive assessment provided. We carefully revised the entire text to remove unnecessary articles and shorten longer sentences. We believe this has significantly improved the manuscript's readability.

Answers to reviewer #1

Answers to reviewers' comments related to taxonomy assignment gave clarity about the focus of the "core microbiome" definition in this work.

Answers to reviewer #2

Compared to the first paper, now the logic structure in the manuscript is clearer. The terms of "true or "accurate" core were clarified. The extra analysis performed to address concerns about meaningful differences in core membership, give evidence and statistical support to show the reader the significant differences when using each method.

Could you change the names you used to classify the core membership assignment methods? Instead of summation of sequence reads, you can use "Abundance" and instead of proportion of replicates, you can use "Occurrence". For methods including both you can name it "Abundance-occurrence". This way will be easier to compare with other papers discussion about the core membership assignment methods.

This is a great suggestion. We have changed the names of the core assignment methods accordingly.

Reviewer #4 (Comments for the Author):

I was not one of the original reviewers, but I have read the revised manuscript and the author responses to the original comments. In general, I agree with the previous reviewers about the expected utility of the R package for microbiome scientists who want to explore different core definitions on their datasets, and with the utility of simulations in better exploring core definitions. I also agree with the previous reviewers' sentiment that the readers may be left with a "now what?" feeling at the end of the work, as the authors conclude that these various definitions may not be useful. Ultimately, I assess that the authors have addressed the original major comments while leaving open to the readers to decide whether and how they want to apply a core definition based on their study's scientific question.

I offer two additional comments (which do not require additional analysis).

First, the authors describe the importance of the separate factors of abundance and prevalence. There was a 2019 piece in Current Opinion in Microbiology that directly considered both types of factors in determining a core and then also partitioned the contribution of the core to the beta diversity explained

by the "core" subset as compared the whole community. This prior piece has a similar philosophy as the current piece. While the prior work is referenced among other review-type articles early in the introduction, it is not discussed in the context of this new study. However, the 2019 piece introduces the idea that the contribution of the core members to beta diversity is a metric of the "success" of the core definition applied to a study, as core members would ultimately be the taxa driving the major ecological patterns, which is now reinforced in the current study. In this sense, the 2019 piece can be used as precedent to motivate the consideration of the core contribution to beta diversity as an output of ecological insight. This discussion of the 2019 reference would potentially fit into the paragraph starting line 339 about predictors of beta diversity.

Reference: Shade, A., & Stopnisek, N. (2019). Abundance-occupancy distributions to prioritize plant core microbiome membership. *Current opinion in microbiology*, 49, 50-58.

Thank you for this suggestion. We have now included the Shade & Stopnisek manuscript in our discussion. We included it to introduce the importance of core membership for patterns of beta-diversity and suggest that their framework can be further used to ad-hoc test the validity of core microbiome assignments. Please, see lines 338-345 and 435-441.

Second, in the simulations the authors vary magnitude of difference and the variance of the abundances of the taxa in the simulated communities and define their "true core" by 1x- 25x differences in the abundance. If I understand this correctly, all true core taxa will have higher abundance than non-core taxa in the simulations. In addition, the prevalence parameter is considered as introduction of noise (precision) across replicates (as an outcome of potential incomplete observation?) and not as directly as their distribution e.g. over space or time/across samples collected in a study. Thus, the simulation seems to bias towards the abundance factor in its exploration of core definitions and uses a replicate-based definition of prevalence/precision rather than a dataset-wide definition. Given this, perhaps it is not surprising that the results confirm the importance of the differential abundances between true core and non-core (e.g., Line 282 "In general, we found a large difference in the abundance of core and non-core taxa ($\pi_{\text{core}}/\pi_{\text{non-core}}$, with varying degrees of precision) to lead to greater accuracy in the identification of core taxa". This point - that the simulations create that core taxa by definition have higher abundance than non-core - should be elevated and discussed so that readers can more clearly understand and evaluate. This comment echoes a previous reviewer's request to clarify the simulations to "lay" readers so that they can understand the assumptions and their consequences.

Yes, this is a correct interpretation of our simulation models. Indeed, all true core members in our simulations will have higher $\pi_{\text{core}}/\pi_{\text{non-core}}$ than the non-core counterparts. However, the precision parameter, especially at lower $\pi_{\text{core}}/\pi_{\text{non-core}}$ (e.g., 1, 2, and 5) incorporates variation in $\pi_{\text{core}}/\pi_{\text{non-core}}$, thus allowing non-core taxa to have higher abundances than their core counterparts. The importance of the precision parameter (θ) (along with differences in abundances) is shown in figure 3 with variation in core assignment accuracy being seen on both the abundance and precision axis. We now discuss the importance of both abundance and prevalence in the main text (see lines 368-373). *"For example, despite the overall higher accuracy of occupancy-based methods as validated by the simulation models, these methods produced inaccurate assignments of core taxa in datasets displaying only small differences in abundance between core and non-core taxa, often leading to an overinflation of the core size based on erroneous assignments. This pattern indicates that both the abundance and prevalence of taxa is important for accurately determining core membership in our simulations, even though all true core taxa were simulated to have higher abundances than their non-core counterparts."*

January 4, 2023

Dr. Gordon Fritz Custer
The Pennsylvania State University
University Park, PA 16802

Re: mSystems01066-22R1 (Comparative analysis of core microbiome assignments: implications for ecological synthesis)

Dear Dr. Gordon Fritz Custer:

Thank you for your thoughtful contribution to mSystems!

Your manuscript has been accepted, and I am forwarding it to the ASM Journals Department for publication. For your reference, ASM Journals' address is given below. Before it can be scheduled for publication, your manuscript will be checked by the mSystems production staff to make sure that all elements meet the technical requirements for publication. They will contact you if anything needs to be revised before copyediting and production can begin. Otherwise, you will be notified when your proofs are ready to be viewed.

Publication Fees:

If you would like to submit a potential Featured Image, please email a file and a short legend to mSystems@asmusa.org. Please note that we can only consider images that (i) the authors created or own and (ii) have not been previously published. By submitting, you agree that the image can be used under the same terms as the published article. File requirements: square dimensions (4" x 4"), 300 dpi resolution, RGB colorspace, TIF file format.

We recognize that the video files can become quite large, and so to avoid quality loss ASM suggests sending the video file via <https://www.wetransfer.com/>. When you have a final version of the video and the still ready to share, please send it to mSystems staff at mSystems@asmusa.org.

Sincerely,

Ashley Shade
Editor, mSystems

Journals Department
E-mail: mSystems@asmusa.org